# Differential Role of Aldosterone and Transforming Growth Factor Beta-1 in Cardiac Remodeling

**DOI:** 10.3390/ijms241512237

**Published:** 2023-07-31

**Authors:** Piotr Kmieć, Stephan Rosenkranz, Margarete Odenthal, Evren Caglayan

**Affiliations:** 1Department of Endocrinology and Internal Medicine, Medical University of Gdańsk, 80214 Gdańsk, Poland; piotr.kmiec@gumed.edu.pl; 2Clinic for Internal Medicine III and Cologne Cardiovascular Research Center, Cologne University Heart Center, 50937 Köln, Germany; stephan.rosenkranz@uk-koeln.de; 3Institute of Pathology, University Hospital of Cologne and Center for Molecular Medicine, University of Cologne, 50937 Köln, Germany; m.odenthal@uni-koeln.de; 4Department of Cardiology, University-Medicine Rostock, 18057 Rostock, Germany

**Keywords:** k aldosterone, transforming growth factor beta-1, ventricular cardiac remodeling, transgenic mice, mitogen-activated protein kinases

## Abstract

Angiotensin II, a major culprit in cardiovascular disease, activates mediators that are also involved in pathological cardiac remodeling. In this context, we aimed at investigating the effects of two of them: aldosterone (Ald) and transforming growth factor beta-1 (TGF-β1) in an in vivo model. Six-week-old male wild-type (WT) and TGF-β1-overexpressing transgenic (TGF-β1-TG) mice were infused with subhypertensive doses of Ald for 2 weeks and/or treated orally with eplerenone from postnatal day 21. Thehearts’ ventricles were examined by morphometry, immunoblotting to assess the intracellular signaling pathways and RT qPCR to determine hypertrophy and fibrosis marker genes. The TGF-β1-TG mice spontaneously developed cardiac hypertrophy and interstitial fibrosis and exhibited a higher baseline phosphorylation of p44/42 and p38 kinases, fibronectin and ANP mRNA expression. Ald induced a comparable increase in the ventricular-heart-weight-to-body-weight ratio and cardiomyocyte diameter in both strains, but a less pronounced increase in interstitial fibrosis in the transgenic compared to the WT mice (23.6% vs. 80.9%, *p* < 0.005). Ald increased the phosphorylation of p44/42 and p38 in the WT but not the TGF-β1-TG mice. While the eplerenone-enriched chow partially prevented Ald-induced cardiac hypertrophy in both genotypes and interstitial fibrosis in the WT controls, it completely protected against additional fibrosis in transgenic mice. Ald appears to induce cardiac hypertrophy independently of TGF-β1, while in the case of fibrosis, the downstream signaling pathways of these two factors probably converge.

## 1. Introduction

Pathological cardiac remodeling (CR) comprises structural and functional changes in the heart due to injury and/or chronic diseases. Ventricular hypertrophy and fibrosis are two major alterations seen in CR. While hypertrophy may serve adaptive purposes at least initially and/or in some cases, it has been established as a risk factor for cardiovascular (CV) events. Pathological cardiac hypertrophy is accompanied by a fibrotic response, which triggers both systolic and diastolic dysfunction, leading to heart failure [1]. Both aldosterone (Ald) and transforming growth factor beta-1 (TGF-β1) have been implicated in the development of hypertrophy and fibrosis in the course of CR [2,3]. 

Ald is a member of the renin–angiotensin–aldosterone system (RAAS) that regulates blood pressure and fluid balance by increasing sodium and water reabsorption via its mineralocorticoid receptor (MR) present in the renal tubules. Apart from this classic effect, Ald directly influences the CV system: MRs are present in vascular smooth muscle cells, fibroblasts and cardiomyocytes, among others. The adverse effects of Ald include cardiac hypertrophy and fibrosis, endothelial dysfunction and inflammation [2]. Recent research in humans indicates that the prevalence of primary aldosteronism as a cause of hypertension may be as high as 20–30%, but it remains largely undiagnosed. This is unsatisfactory since patients with primary aldosteronism suffer from much higher CV mortality and morbidity than those with primary hypertension, despite the availability of anti-MR targeted pharmacotherapy [4,5,6].

TGF-β1 is recognized as a fundamental mediator of fibrotic processes in kidney, lung, liver and heart diseases [7]. Elevated TGF-β1 expression has been shown in idiopathic hypertrophic cardiomyopathy, dilatative myopathy and the transition from stable hypertrophy to heart failure. At the molecular level, TGF-β1 acts as a local, para- and autocrine mediator as well as a secondary messenger in response to angiotensin II, Ald and beta-adrenergic signaling [3,8,9,10,11,12]. Upon binding to its receptor, TGF-β1′s intracellular signal is transduced via Smad transcriptional activators; additionally, mitogen-activated protein kinases (MAPKs), i.e., the Erk, p38 and c-Jun N-terminal kinase, are activated through TGF-β1-activated kinase 1.

In this study, we analyzed the individual effects of Ald and TGF-β1 on cardiac hypertrophy and fibrosis as well as the downstream signaling pathways in a murine model.

## 2. Results

### 2.1. Baseline Characteristics of TGF-β1 Mice

At baseline, the mice with a constitutive expression of the TGF-β1 transgene exhibited cardiac hypertrophy, as indicated by a higher mean biventricular heart weight (vHW) to body weight (BW) ratio (vHW/BW)(+28.6%, *p* < 10^−4^) and cardiomyocyte diameter (+24.2%, *p* < 10^−4^) compared to the WT strain (Table 1 and Figure 1A–C). Furthermore, the atrial natriuretic peptide (ANP) mRNA expression was 89.7% higher in the TGF-β1-overexpressing transgenic mice (TGF-β1-TG or TG) compared to the wild-type (WT) mice, *p* = 0.041 (Figure 1D).

Concerning cardiac fibrosis, the area occupied by the interstitial fibrous tissue in the myocardium of the TG animals was almost twice that of the WT mice: 1.03 ± 0.2 vs. 1.91 ± 0.27%, *p* < 10^−4^ (Figure 2A,C). Moreover, the TGF-β1-TG mice showed a trend toward higher fibronectin (FBN) mRNA expression in the ventricular tissue compared to the WT animals (*p* = 0.15) (Figure 2D). In contrast, the perivascular fibrosis score was similar for both strains (Figure 2B).

A constitutive higher Erk (p44/42) and p38 kinase phosphorylation in the ventricular heart tissue samples from the TGF-β1-TG compared to the WT mice was recorded: 7.8 (5.9–9.6)- and 11 (5.2–17.8)-fold, respectively (Figure 3A).

### 2.2. Effects of Ald and Ep on Cardiac Hypertrophy

The 14-day Ald infusion led to a 9.8 ± 2.5% increase in the vHW/BW in the WT mice (*p* < 10^−4^), while in the TGF-β1-TG animals, the increase was insignificant (*p* = 0.31), and the change in the vHW/BW due to Ald was comparable between strains at *p* = 0.57 (Figure 4A). The increase in the cardiomyocyte diameter was more pronounced in the WT than TG mice: 36 ± 13% versus 24.6 ± 2.7%, *p* = 0.026 (Figure 4B). In contrast, the ANP mRNA expression did not differ between the respective untreated and Ald-infused mice of each strain (Figure 1D).

Under basal conditions, eplerenone (Ep) had no effect on the vHW/BW or cardiomyocyte size in either mouse strain (Figure 1A–C). Accordingly, no differences were recorded in ANP mRNA expression between the control and Ep-enriched-chow groups (Figure 1D). 

In the Ald-infused animals, the vHW/BW was not significantly affected by the concurrent administration of Ald and its receptor antagonist in mice of either genotype, although a trend toward hypertrophy attenuation could be observed in the TGF-β1-TG animals (Figure 1A). The Ep administration with a concurrent Ald infusion led to a 13.5% decrease in the cardiomyocyte diameter in the WT mice versus those only treated with Ald, but not in the TGF-β1-overexpressing mice (Figure 1B). No significant differences in the ANP mRNA expression were found between the groups infused with Ald only and those treated with both Ald and Ep (Figure 1D). 

### 2.3. Effects of Ald and Ep on Cardiac Fibrosis

While both strains exhibited more interstitial fibrosis in the ventricles upon completion of the two-week Ald infusion, the increase was substantially higher in the WT mice: 80.9 ± 32% versus 23.6 ± 30%, *p* = 0.002 (Figure 4C). These morphometric observations were supported at the molecular level, as Ald induced a significant increase (+47.7%) in cardiac FBN mRNA expression only in the WT mice (*p* = 0.029, Figure 2D).

Importantly, the increase in cardiac interstitial fibrosis recorded in the WT strain was significantly higher than the increases in the vHW/BW and cardiomyocyte diameter (ANOVA, *p* < 0.0001, Figure 4). This was not the case in the transgenic animals (*p* = 0.175). 

As expected, the Ep treatment itself had no gross morphological effect on cardiac interstitial and perivascular fibrosis in either the WT or TGF-β1-TG mice (Figure 2A–C). However, a trend was recorded toward a lower myocardial expression of FBN mRNA in the Ep-treated groups compared to their respective controls of both genotypes (Figure 2D). 

Further, Ep effectively decreased Ald-induced cardiac interstitial fibrosis in the TGF-β1-TG mice, i.e., by 28% (TG + Ald + Ep versus TG + Ald). A similar trend was seen in the WT animals, although it failed to reach statistical significance (Figure 2A). Importantly, the Ep treatment nearly completely abolished the Ald-induced upregulation of the FBN mRNA in both the WT and TGF-β1-TG mice (Figure 2D). These results indicate that Ald-induced cardiac interstitial fibrosis is mainly mediated by the MR and can be effectively attenuated by receptor antagonism.

### 2.4. Effects of Ald and Ep on MAPK Signaling

Since MAPK signaling is an important pathway mediating cardiac hypertrophy, we analyzed the phosphorylation of the main MAPKs, Erk (p44/42) and p38 in the ventricular heart tissues of the WT and TGF-β1-TG mice. In the WT animals, both signaling pathways were activated with Ald infusion (Figure 3B). In the TGF-β1-TG mice, Erk and p38 phosphorylation had already been higher at baseline and did not further increase in response to Ald (Figure 3A,C).

A trend was observed toward the attenuation of Ald-induced Erk (p44/42) and p38 phosphorylation in the ventricular heart tissues of the WT mice as indicated by a significant ANOVA result but insignificant post-hoc comparisons of the four groups (Figure 3B,C). In contrast, Ep had no statistically significant effect on either Erk (p44/42) or p38 phosphorylation in any of the TG groups, suggesting that the activation of these pathways by TGF-β1 occurs independently of the MR.

## 3. Discussion

To our knowledge, this study is the first to examine the effects of continuous Ald infusion and oral Ep treatment on cardiac remodeling in transgenic mice overexpressing TGF-β1. Therefore, direct comparisons with previous research are not possible. 

Our main finding is that the effect of Ald on cardiac hypertrophy is additive to that of TGF-β1, while with respect to interstitial fibrosis, the downstream signaling mechanisms of these two mediators likely converge. Possible explanations include: the enhancement of profibrotic genes such as connective tissue growth factor by both molecules [13,14], immune cell modulation by TGF-β1 antagonizing Ald’s actions [15,16], Ald-mediated osteopontin upregulation that inhibits the antifibrotic TGF-β1 activation of the β2-adrenergic receptor [17,18], interactions within the RAAS system and also an inhibitory effect of excess TGF-β1 on Ald secretion [19]. 

The differential effects on CR can be deduced by comparing the WT and TG strains: in the former, interstitial fibrosis increased significantly more than the cardiac hypertrophy parameters in response to Ald infusion, whereas the change in the morphometric parameters was comparable in the setting of excess TGF-β1 (Figure 4 and Figure 5). Furthermore, the Ep treatment alleviated the profibrotic effect of the Ald infusion in the TG animals, while only a partial reduction was observed for the cardiomyocyte diameter (Figure 5). Concerning the Erk and p38 activation, no significant alterations were observed in the case of the TGF-β1-overexpressing mice across all groups: control, Ald and/or Ep treated, while in the WT animals, MAPKs were stimulated by Ald. 

Previous studies concerning the effects of both mediators on CR are scarce. Both molecules were investigated in vivo in mice with a graded TGF-β1 expression to reveal that its excess inhibited adrenal steroid synthesis, including Ald. However, the cardiac morphology was not reported [19]. Conversely, in a study by Nishioka et al., a four-week Ald infusion along with 1% NaCl drinking water induced TGF-β1 mRNA expression and marked cardiac fibrosis [20]. Next, Ald stimulated TGF-β1 expression in cardiac fibroblasts and rat mesangial cells [21,22]. Still, a difficulty arises in discriminating the effect of Ald and Ang II on downstream signal transduction via TGF-β1 in light of interactions within the RAAS [23,24,25,26]. While Ang II has been established as a TGF-β1-inducing molecule, its antagonism has not been effective at abolishing TGF-β1’s cardiac effects [27]. 

A larger body of data has been generated with respect to TGF-β1’s role in cardiac remodeling alone. Previous studies using the same transgenic model also revealed cardiac hypertrophy and fibrosis [28,29,30]. Methodological differences and the possible less homogenous expression of the TGF-β1 transgene years after the cited reports (apparent in greater variation in the vHW/BW) may account for lower hypertrophy indices, while findings concerning ANP mRNA expression and interstitial fibrosis are comparable [29,30,31]. Data from studies with other animal models echo ours. Nakajima et al. showed that TGF-β1 overexpression increased heart-weight-to-tibia-length ratios as well as cardiomyocyte size, and although a profibrotic effect was seen only in the atria, this discrepancy had been partially explained by the distinct approaches to TGF-β1 overexpression as well as the low activity of the factor [29,30,32]. Further, TGF-β1 −/− and Rag1 −/− double-knockout mice did not develop cardiac hypertrophy induced by subpressor Ang II doses, as was the case for the controls [10]. Moreover, age-associated myocardial fibrosis was ameliorated in single-knockout TGF-β1 +/− mutant mice [33]. Finally, a TGF-β1 neutralizing antibody prevented diastolic dysfunction in pressure-overloaded rats [34]. 

Analogically to morphometric findings, ample research exists supporting higher MAPK phosphorylation due to TGF-β1, which induces both Erk 1/2 [35,36,37] and p38 phosphorylation (particularly via the noncanonical, i.e., non-Smad-related, TGF-β1-activated kinase 1 pathway) [3,38,39]. Evidence has been accumulating for the role of p38 in the development of cardiac fibrosis and Erk primarily in hypertrophy [40,41,42,43]. 

In discussing the effects of Ald on CR, the current study contrasts with most preceding ones in that neither nephrectomy nor salt loading was applied. Also, here, Ald was infused at doses known not to induce hypertension (although a lack of blood pressure measurements is a limitation of our data). In this setting, Ald’s actions could be extracted from the influence of other factors. With respect to the MR antagonist, Ep was initiated three weeks prior to the Ald infusion, while other researchers have applied simultaneous treatments. 

Bearing these in mind, in a study similar to ours, Yoshida et al. applied an Ald infusion at a rate of 0.75 µg/h for 14 days in male Sprague Dawley rats. A minor blood pressure increase was successfully reduced by an antioxidant without preventing cardiac hypertrophy (revealed by echocardiography and a 34% increase in the cardiomyocyte cross-sectional area). Spironolactone effectively attenuated Ald-induced cardiac hypertrophy [44]. A similar methodology, yet with 1% NaCl instead of drinking water, was applied by Nakano et al. to demonstrate a higher LV weight/BW ratio with Ald infusion and a prevention thereof by spironolactone [45]. In another model, Ep ameliorated the cardiac hypertrophy and fibrosis exhibited by mice overexpressing 11-beta-hydroxysteroid dehydrogenase 2 (which promotes Ald-MR binding) [46]. In contrast, Iglarz et al. did not record macroscopic cardiac hypertrophy after a 6-week Ald infusion in rats [47]. 

As for cardiac interstitial fibrosis, an approximate 80% increase and higher FBN mRNA expression were recorded here in the Ald-infused WT mice compared to the untreated controls, which is in line with previous studies: that of Iglarz et al. mentioned above and another: by Johar et al., who also demonstrated a higher FBN expression in Ald-infused rodents [47,48]. Both Ep and spironolactone were effective at preventing fibrosis [20,49,50].

Prior in vitro research also supports the trend toward higher Erk and p38 phosphorylation in response to Ald recorded here, e.g., [51]. In vivo data are scarce, yet the results by Nakano et al. mentioned above include this observation for Erk [45].

In the current study, Ep only partially prevented Ald-induced cardiac hypotrophy (there was a significant difference in the cardiomyocyte diameter, but not the vHW/BW) and interstitial fibrosis (the WT + Ald + Ep group was not significantly different from the WT and WT + Ald groups) (Figure 5), which contrasts with other reports [44,45,46,52]. Possibly, compensatory mechanisms to the Ep treatment developed in postnatal weeks 3–6, which rendered MR antagonism ineffective once Ald was administered in weeks 6–8. These might include a higher MR expression and Ald secretion, as well as altered Ang II signaling. The hypothesis of Ald escape is supported further by the data on the kidney-weight-to-BW ratios from the ‘+Ald + Ep’ group, which were higher than those of the untreated controls (Table 1). 

## 4. Materials and Methods

### 4.1. Treatment of Animals

Male WT, i.e., C57Bl/J6 (Harlan and Winkelmann), and TGF-β1-TG were used in this study. 

The TGF-β1-TG mice were originally generated by Sanderson et al. by using a murine albumin promoter and enhancer linked to a porcine TGF-β1 construct and the 3′ region of the human growth hormone gene, containing a polyadenylation signal [28]. The preferential secretion of mature TGF-β1 resulted from cysteine/serine substitutions at amino acid residues 223 and 225 in the TGF-β1 c-DNA. Line 25 mice were used here, in which solely the males were transgenic and exhibited a 10-fold increased plasma TGF-β1 concentration compared with their age-matched controls [53]. The line was maintained by continued backcrosses to the WT F1 mice (which served as ourcontrols). 

All the investigations were performed in accordance with the National Institute of Health’s ‘Guide for the Care and Use of Laboratory Animals’ and Institutional Animal Care and Use Guidelines.

Eight study groups (7 to 14 mice per each group) were investigated, i.e., four WT and four TGF-β1-TG groups: controls with a regular chow diet and without an Ald infusion (WT, TG), with an Ald infusion (WT + Ald, TG + Ald), with an Ep-enriched chow diet (WT + Ep, TG + Ep) and both fed an Ep-enriched chow dietand with an Ald infusion (WT + Ald + Ep, TG + Ald + Ep). 

The selective MR antagonist Ep was provided from Pfizer in powder form and added to mouse chow by Harlan Laboratories (1.5 mg of Ep per gram of chow). A daily dosage of approximately 200–300 mg per kg BW was obtained. Ald (ordered from Sigma) was dissolved in 20% ethanol to a concentration of 2 mg/mL and stored at 4 °C. Dilution with 0.9% NaCl to a final 1 mg/mL concentration took place immediately prior to filling the solution into miniature infusion pumps (Alzet, model 1002). The manipulations were performed in sterile conditions. 

The treatment with Ep started immediately after weaning (at 3 weeks of age) by providing the drug-enriched chow, which lasted 5 weeks (until the age of 8 weeks), while Ald was infused via a minipump between day 42 and 56 (Figure 6).

Infusion pumps with Ald were implanted subcutaneously at the age of 6 weeks during isoflurane anesthesia. A total of 100 microliters of the Ald solution was infused continuously for 14 days with an appropriate flow moderator, equivalent to a circa 0.35 mg/kg BW/day dose, i.e., a subhypertensive one [54]. Three control mice received pumps containing 0.9% NaCl to verify whether the pump implantation had any effect on their cardiac morphology, which was not the case. 

### 4.2. Organ Extraction and Sample Preparation

At the age of 8 weeks, the animals’ organs were collected. Following isoflurane anesthesia, their BW was measuredthe mice were sacrificed by cervical dislocation, their hearts were excised and perfused with cold saline to remove the blood. The vHW was recorded after excising the atria. The hearts were sliced to obtain samples for: immunohistochemistry (the heart base was cut off at the organ’s equator and stored in a 4% formaldehyde solution overnight), immunoblotting (the heart was snap frozen with liquid nitrogen and stored at −80 °C) and mRNA quantification (the RNA-Later-containing tube was stored at 4 °C). Had a minipump been implanted, it was excised and weighed after the organ extraction, and the BW of the mouse was reduced accordingly.

### 4.3. Morphometric Analysis

Cardiac hypertrophy was assessed by determining the vHW/BW (in milligrams per gram) as well as the mean cardiomyocyte diameter. The latter was acquired by measuring at least 100 cell diameters per animal. Horizontal sections at the equator of the heart were stained with hematoxylin-eosin. Ten sections were analyzed for each mouse. For each section, the diameter from at least ten cardiomyocytes was determined and the averages were recorded. To ensure consistency, only the cells with nuclei at their circumference were chosen. Here, the shortest myocyte diameter was measured.

The midventricular sections were stained with Masson’s trichrome to evaluate the amount of fibrous tissue in the muscle. Cardiac interstitial fibrosis was quantified in 9 to 12 randomly chosen areas for each mouse heart (from a single heart section) in a blinded manner. A 20× magnification was used. The regions with blood vessels as well as pericardium or endocardium were omitted. For every chosen area, the percentage of fibrous tissue was computed by using a color analysis method with Olympus’ Cell^P 5.0 software. Blue- and grey-stained fibrotic elements were discriminated from white, artefactual, intracellular areas and black nuclei by setting color thresholds. The white artefacts were subtracted from the frame area set for each region to acquire an accurate percentage of the fibrotic material. Perivascular fibrosis was assessed in a semiquantitative way of randomly labeled Masson’s trichrome stains. Coronary vessels were photographed—at least 5 for each animal—and scored on a 0 to 4 scale (with “0” representing no fibrosis, “1” minimal and “4” massive fibrosis). The average cardiac perivascular fibrosis score was calculated for each mouse.

### 4.4. Western Blotting

The following antibodies were provided by Cell Signaling technology: the phospho-p44/42 MAP kinase (Thr202/Tyr204) antibody (product number 9101), p44/42 MAP kinase antibody (product number 9102), phospho-p38 MAP kinase (Thr180/Tyr182) antibody (product number 9211) and p38 MAP kinase antibody (product number 9212). The antirabbit monoclonal antibody was supplied by Sigma. The 69.3 anti-Ras GTP activating protein (RasGap) antibody was kindly supplied by professor A. Kazlauskas from Harvard Medical School [55].

Ventricular heart lysates were prepared from liquid-nitrogen snap-frozen pieces by homogenization in a 1 mL RIPA proteinase inhibiting buffer at 4 °C. Subsequently, the homogenates were incubated for 2 h at 4 °C while experiencing gentle agitation and were next centrifuged for 20 min at 10,000 g. The supernatant was collected, aliquoted and stored at −80 °C. For Western blotting, the samples were thawed on ice. The protein concentration was quantified by NanoDrop (NanoDrop Technologies, Inc., Wilmington, DA, USA) and ranged from 14 to 27 mg/mL. The aliquots were diluted with RIPA to acquire equal sample concentrations. The homogenates were suspended in a 4 × SDS sample buffer. Samples with equal protein amounts were run on SDS–PAGE and transferred to PVDF membranes. The blots were probed with appropriate antibodies to reveal protein bands on the hyperfilm, which were quantified by using ImageJ 1.54d software as described previously [56].

### 4.5. Real-Time qPCR

The cardiac expression of the mRNAs of two marker genes was investigated: ANP for cardiac hypertrophy and FBN for fibrosis. 

Rodent glyceraldehyde 3-phosphate dehydrogenase (GAPDH) primers and a probe with a VIC reporter dye were supplied by Applied Biosystems (product number 4308313). The ANP and FBN primers and FAM reporter dye probes were ordered from Eurofins MWG Operon: ANP probe 5′-TCGCTGGCCCTCGGAGCCTAC-3′, forward primer 5′-GAAAAGCAAACTGAGGGCTCTG-3′, reverse primer 5′-CCCCGAAGCAGCTGGAT TGC-3′, FBN probe 5′-TCGGAGCCATTTGTTCCTGCACGT-3′, forward primer 5′-TGT-AGGAGAACAGTGGCAGAAAGA-3′ and reverse primer 5′-CCGCTGGCCTCCGAA-3′.

The total RNA was extracted from the heart tissues stored in RNA Later by using the TRIzol (Invitrogen, Carlsbad, CA, USA) RNA extraction method [57]. Equal amounts of the isolated RNA were subsequently transcribed into cDNA by using a high-capacity cDNA reverse transcription kit (Applied Biosystems, Waltham, MA, USA) as per the manufacturer’s instructions. The iQ SYBR Green Supermix (Bio-Rad, Hercules, CA, USA) kit was applied to perform a qPCR. The mRNA expression was analyzed via the ΔCt method. The Ct values of the target genes (ANP and FBN) were normalized to that of GAPDH (reference gene) by using the equation ΔCt = Ct(reference) − Ct(target) and were expressed as ΔCt. 

### 4.6. Statistical Analysis

The control WT and TG animals were tested for differences with a Student’s *t*-test or its nonparametric variant, the Mann–Whitney U test, depending on the data distribution, which was verified with the Shapiro–Wilk test. For each strain, the parameters in the four groups (control, Ald infused, Ald infused and Ep treated and Ep treated) were tested for differences by using parametric (with Tukey’s post-hoc test) or nonparametric (Kruskal–Wallis) ANOVAs depending on the data distribution. The results are presented as the mean ± standard deviation (SD) or median (interquartile range, IQR), and the graphs present the mean and SD values. Statistical significance was set at 0.05. 

## 5. Conclusions

In the current and previous studies, TGF-β1 has been shown to be a vital cardiac pro-fibrotic and prohypertrophic mediator. The data obtained here also suggest that in the myocardium, the profibrotic actions of TGF-β1 and Ald overlap, while they are at least in part divergent in the case of cardiac hypertrophy. Further research is necessary to elucidate the cross talk between all components of the RAAS and TGF-β1.

## Figures and Tables

**Figure 1 ijms-24-12237-f001:**
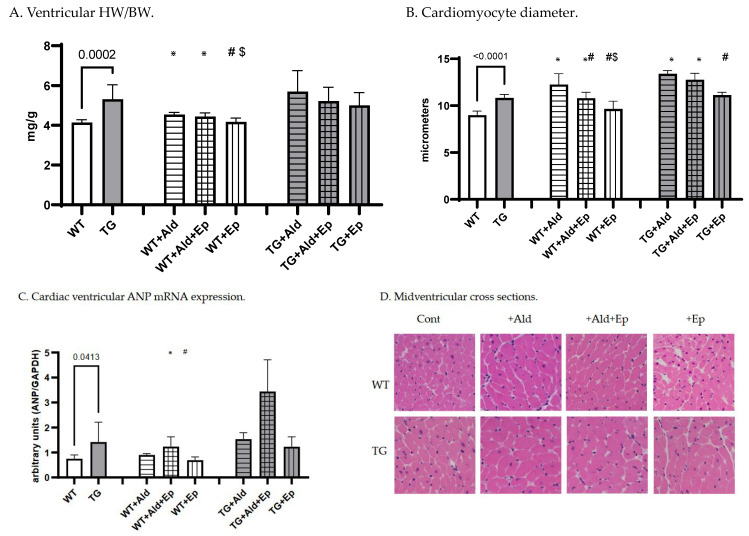
Cardiac hypertrophy morphometric and ANP mRNA expression data. Significance level for the WT vs TG comparison is given; symbols annotate respective comparisons with control (*), ‘+Ald’ (#) and ‘+Ald + Ep’ ($) groups. (**A**)—ventricular heart weight/body weight; (**B**)—mean diameters of ventricular cardiomyocytes; (**C**)—ANP mRNA expression in ventricular heart tissues normalized to GAPDH; (**D**)—representative microphotographs of midventricular cross sections. At the same magnification (200×), differences in cardiomyocytes’ diameters can be noticed. HE staining.

**Figure 2 ijms-24-12237-f002:**
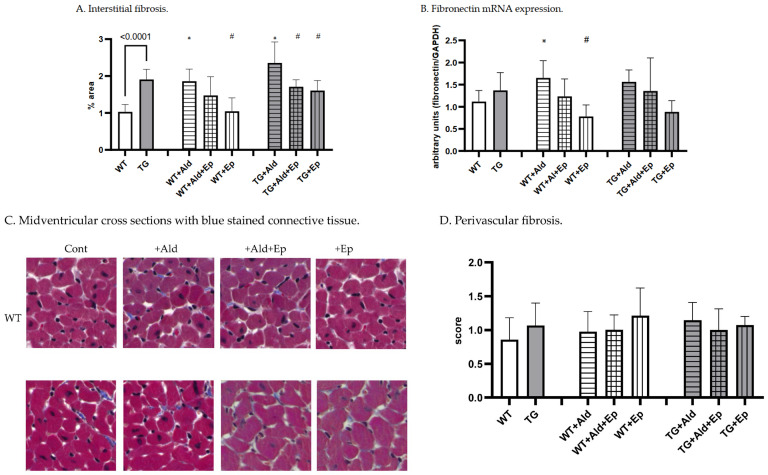
Cardiac fibrosis morphometric and fibronectin mRNA expression data. Significance level for the WT vs TG comparison is given; symbols annotate respective comparisons with control (*), ‘+Ald’ (#). (**A**)—cardiac interstitial fibrosis under basal and experimental conditions; (**B**)—fibronectin mRNA expression in ventricular heart tissues normalized to GAPDH mRNA; (**C**)—representative microphotographs, whereby at the same magnification (400×), the amount of fibrotic material can be noticed. Masson’s trichrome staining: (**D**)—a scoring system was used to assess perivascular fibrosis as described in ‘Methods’.

**Figure 3 ijms-24-12237-f003:**
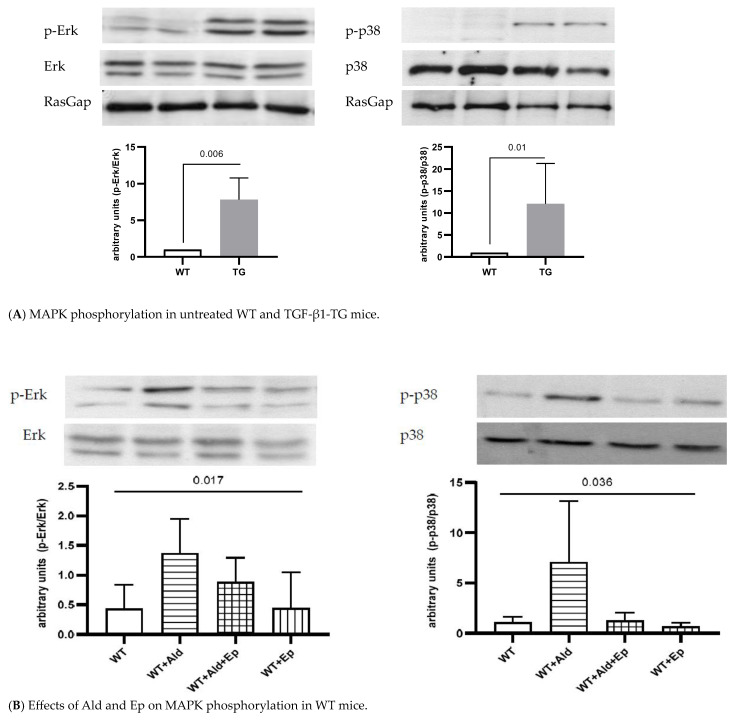
Western blot analysis. Representative Western blot scans of *p*-Erk and *p*-p38 of WT and TG animals with total Erk and p38 loading control. (**A**)—control (untreated) WT and TG animals; (**B**,**C**)—untreated, Ald-infused (‘+Ald’), Ald-infused and Ep-treated (‘+Al + Ep) eplerenone-treated (‘+Ep) wild-type (WT) and TGF-β1-transgenic (TG) mice. Four or five separate experiments were performed for each strain. Significance level for ANOVAs on WT mice is given, and post-hoc comparisons between groups were insignificant. . RasGap—Ras GTP activating protein.

**Figure 4 ijms-24-12237-f004:**
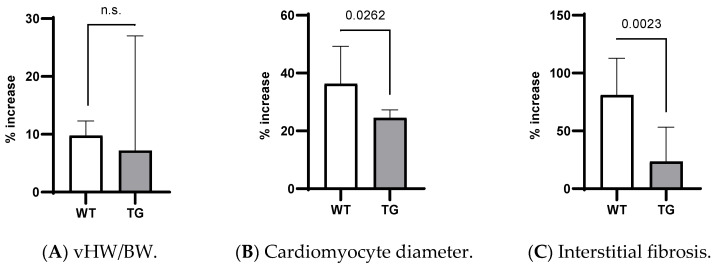
Relative changes in morphometric parameters due to Ald infusion in WT and TG mice: (**A**) vHW/BW, (**B**) cardiomyocyte diameter, (**C**) interstitial fibrosis. n.s.—not significant; TG—TGF-β1 transgenic; vHW/BW—ventricular-heart-weight-to-body-weight ratio; WT—wild type.

**Figure 5 ijms-24-12237-f005:**
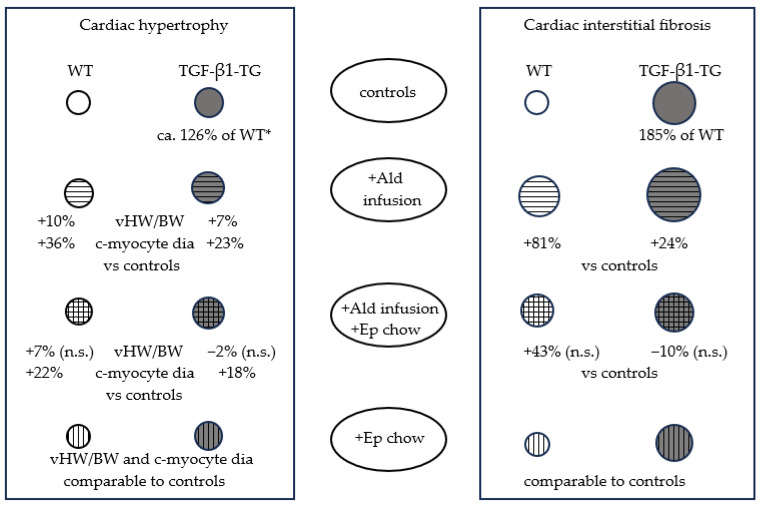
Schematic presentation of morphometric data. Circle diameter for WT controls was used as reference for other WT groups and TGF-β1-TG controls; since two parameters were used for cardiac hypertrophy, the circle diameter reflects an intermediate value between the difference in vHW/BW and cardiomyocyte diameter versus controls. * Mean between difference in vHW/BW and c-myocyte diameter for the TGF-β1-TG vs. WT controls comparison is given. Ald—aldosterone; c-myocyte dia—cardiomyocyte diameter; Ep chow—eplerenone-enriched chow; n.s.—not significant; TGF-β1-TG—TGF-β1-overexpressing transgenic mice; WT—wild-type mice; vHW/BW—ventricular-heart-weight-to-body-weight ratio.

**Figure 6 ijms-24-12237-f006:**
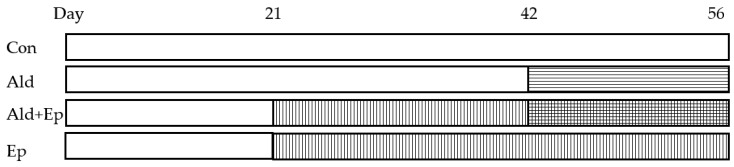
Mice treatment scheme. There were four study groups for both WT and TG mice. Regular or eplerenone-enriched chow was given from day 21 to 56, while Ald was infused subcutaneously via a minipump from day 42 to 56. Ald—aldosterone-infused groups; Ald + Ep—both Ald-infused and Ep-enriched-chow-fed groups; Con—control groups; Ep—eplerenone-enriched-chow-fed groups.

**Table 1 ijms-24-12237-t001:** Morphometric and qPCR data on wild-type and TGF-β_1_-TG mice.

		WT	TG	*p*	WT + Ald	WT + Ald + Ep	WT + Ep	*p* (WT)	TG + Ald	TG + Ald + Ep	TG + Ep	*p* (TG)
**Ventricular HW** **/BW**	*n*	8	14		7	7	8		9	9	9	
	mg/g	4.1 ± 0.1	5.3 ± 0.7	<10^−3^	4.5 ± 0.1 *	4.4 ± 0.2 *	4.2 ± 0.2 ^#$^	<10^−4^	5.7 ± 1.1	5.2 ± 0.7	5 ± 0.6	0.31
Cardiomyocyte diameter	*n*	8	9		7	7	8		9	9	9	
	µm	8.8 (0.8)	10.9 (0.3)	<10^−4^	12.3 ± 1.2 *	10.8 ± 0.6 *^#^	9.6 ± 0.8 ^#$^	<10^−4^	13.4 (0.5) *	12.8 (1.1) *	11 (0.5) ^#^	<10^−4^
ANP/GAPDH mRNA	*n*	7	14		6	7	6		7	7	6	
	-	0.8 ± 0.2	1.4 ± 0.8	0.041	0.9 ± 0.06	1.2 ± 0.4 *	0.7 ± 0.1 ^$^	10^−3^	1.5 ± 0.3	3.4 ± 1.3 *^#^	1.2 ± 0.4 ^$^	<10^−4^
Interstitial fibrosis	*n*	8	9		7	7	8		9	9	10	
	%	1 ± 0.1	1.9 ± 0.1	<10^−4^	1.9 ± 0.1 *	1.5 ± 0.2	1.05 ± 0.1 ^#^	<10^−3^	2.4 ± 0.2 *	1.7 ± 0.1 ^#^	1.61 ± 0.1 ^#^	<10^−3^
Perivascular fibrosis	*n*	8	9		7	6	6		9	9	10	
	score	0.9 ± 0.3	1.1 ± 0.3	0.21	1 ± 0.3	1 ± 0.22	1.2 ± 0.4	0.27	1.2 ± 0.26	1 ± 0.3	1.1 ± 0.1	0.73
FBN/GAPDH mRNA	*n*	7	15		7	7	7		7	9	7	
	-	1.1 ± 0.3	1.4 ± 0.4	0.15	1.7 ± 0.4 *	1.2 ± 0.4	0.8 ± 0.3 ^#^	<10^−3^	1.6 ± 0.3	1.4 ± 0.8	0.9 ± 0.3	0.06
Kidney W /BW	*n*	8	14		7	10	8		9	9	9	
	mg/g	11.9 ± 0.3	12.8 ± 1.6	0.15	14.9 ± 1.6 *	13.7 ± 1.2 *	11.4 ± 1.5 ^#$^	<10^−4^	15.9 ± 1.5 *	13 ± 0.8 ^#^	12.3 ± 1.8 ^#^	<10^−4^

Legend: *p* for comparisons of WT and TG controls and four-group (each genotype) ANOVAs/Kruskal–Wallis test is given; significant differences in Tukey’s/Dunn’s post-hoc tests are denoted with * vs. controls, ^#^ vs. ‘+Ald’ and ^$^ vs. ‘+Ald + Ep’. * For vHW/BW vs. WT + Ald, *p* = 3 × 10^−4^; WT + Ald + Ep, *p* = 6 × 10^−3^. For cardiomyocyte diameter vs. WT + Ald, *p* < 10^−4^; WT + Ald + Ep, *p* = 10^−3^; TG + Ald, *p* < 10^−4^; TG + Ald + Ep, *p* = 0.006. For ANP/GAPDH mRNA vs. WT + Ald + Ep, *p* = 0.004; TG + Ald + Ep, *p* < 10^−4^. For interstitial fibrosis vs. WT + Ald, *p* < 10^−3^; TG + Ald, *p* = 0.0498. For FBN/GAPDH mRNA vs. WT + Ald, *p* = 0.029. For kidney W/BW vs. WT + Ald, *p* < 10^−3^; TG + Ald, *p* < 10^−4^. ^#^ For vHW/BW vs. WT + Ep, *p* = 8 × 10^−4^. For cardiomyocyte diameter vs. WT + Ald + Ep, *p* = 0.01; WT + Ep, *p* < 10^−4^; TG + Ep, *p* = 6 × 10^−4^. For interstitial fibrosis vs. WT + Ep, *p* = 10^−3^; TG + Ald, *p* = 0.003; TG + Ep, *p* = 3 × 10^−4^. For FBN/GAPDH mRNA vs. WT + Ep, *p* = 3 × 10^−4^. For kidney W/BW vs. WT + Ep, *p* < 10^−4^; TG + Ald + Ep, *p* = 10^−3^; TG + Ep, *p* < 10^−4^. ^$^ For vHB/BW vs. WT + Ep, *p* = 0.019. For cardiomyocyte diameter vs. WT + Ep, *p* = 0.049. For ANP/GAPDH mRNA vs. WT + Ep, *p* = 0.002. For kidney W/BW vs. WT + Ep, *p* = 0.003. Ald—aldosterone; BW—body weight; Ep—eplerenone; GAPDH—glyceraldehyde 3-phosphate dehydrogenase; HW—heart weight; TG—transgenic; W—weight; WT—wild type.

## Data Availability

The data presented in this study are available upon request from the corresponding author.

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
