# Peer review of "Differential Role of Aldosterone and Transforming Growth Factor Beta-1 in Cardiac Remodeling"

_ijms, 2023, doi:10.3390/ijms241512237_

Round 1
Reviewer 1 Report
The manuscript by Kmiec et al demonstrated the “differential role of aldosterone and Transforming growth factor beta-1 in cardiac remodeling”. The authors first identified the baseline characteristics of mice including cardiomyocyte diameter, ANP, interstitial fibrosis, FBN, and Kidney. Interstitial fibrosis tissue in the myocardium of TG animals was shown to double to wild-type mice however it is not the case with perivascular fibrosis. However, the authors should address the following concerns.
1. Did the authors measure heart function? Does it change cardiac output, EF, or FS with respective to treatments?
2. Masson trichrome images should be shown the whole cross-sectional area and the zoomed-out picture would give more clarity than just zoomed images.
3. Does authors perform tunnel study on cardiomyocyte? Do treatments affect apoptosis?
4. To study the cardiomyocyte hypertrophy WGA staining cardiomyocytes will give additional information with changes in the myocyte cross-sectional area?
5. A schematic diagram should be provided.
Author Response
Dear Reviewer 1,
we thank you for your insightful remarks concerning our manuscript entitled ‘Differential Role of Aldosterone and Transforming Growth Factor beta-1 in Cardiac Remodelling’.
Please find responses to your comments below.
The revised manuscript contains changes highlighted in yellow. We hope you will find the corrected manuscript suitable for publication in the ‘International Journal of Molecular Sciences’.
Best regards,
Piotr Kmieć
Comment 1) ‘Did the authors measure heart function? (…)’
Response: In our study, heart function was not examined. We agree echocardiographic evaluation may have added important data in the context of cardiac remodelling.
Comment 2) ‘Masson trichrome images should be shown the whole cross-sectional area and the zoomed-out picture would give more clarity than just zoomed images.’
Response: In the revised manuscript, we include modified images to better indicate the differences between groups. However, these representative images are still zoomed-in, which was done on purpose. Initially, we also believed zoomed-out photographs to be preferable, however, the differences were difficult to discern.
We would like to point out that the percentage of fibrotic elements ranged between 1 and 2 % of the image area, upon excluding white artefacts, etc. as explained in ‘Methods’. This makes it difficult to present zoomed-out, yet small-sized images and reflect the differences (e.g. image area still is taken up mostly by cardiomyocytes, while blue and grey elements are rather scarce).
The quantification of interstitial fibrotic tissue was performed in a blind manner (cross-sections of individual animals were coded by a different person than the examiner). Therefore, in Figure 3C we only aimed at including representative zoomed-in areas that roughly correspond to the results reported in Table 1 and Figure 3A for all eight groups.
Comment 3) ‘Do authors perform tunnel study of cardiomyocyte? Do treatments affect apoptosis?’
Response: Assessment of apoptosis had not been planned in our study. The effect of aldosterone and eplerenone on the apoptosis of cardiomyocytes was outside the scope of our project.
Comment 4) ‘To study the cardiomyocyte hypertrophy WGA staining cardiomyocytes will give additional information with changes in the myocyte cross-sectional area.’
Response: Indeed, wheat germ agglutinin staining is useful in the assessment of cardiac hypertrophy. At this point, howevere, we are not able to apply WGA to our tissue samples.
5) ‘A schematic diagram should be provided’.
Response: We thank for this suggestion. In the revised manuscript we included a diagram with cardiac morphometric data (Figure 6), which serves as a useful reference to points raised in the ‘Discussion’.
Reviewer 2 Report
Dear authors, dear editors,
the work on Caglayan represents an exciting and relevant approach in the field of molecular elucidation of cardiovascular remodeling. Even though there is already preliminary work in a similar design or with a similar question, the presentation of the results in this work has been successful. In terms of content, a clear arc is drawn from the introduction to the methods section to the results, so that I would recommend the acceptance of the publication. Only the smallest editing aspects in the text would have to be addressed.
Author Response
Dear Reviewer 2,
we would like to thank you for your kind assessment of our manuscript entitled ‘Differential Role of Aldosterone and Transforming Growth Factor beta-1 in Cardiac Remodelling’.
Below we respond to your comments in a point-by-point manner.
The revised manuscript contains changes highlighted in yellow.
We hope you will still find the corrected manuscript suitable for publication in the ‘International Journal of Molecular Sciences’.
Best regards,
Piotr Kmieć
Comments
1) ‘The work (…) represents an exciting and relevant approach in the field of molecular elucidation of cardiovascular remodeling’
and
2) ‘(…) The presentation of the results in this work has been successful (…) In terms of content, a clear arc is drawn from the introducion to the methods section to the results, so that I would recommend the acceptance of the publication’.
Response: We thank Reviewer 2 for their kind words.
Comment 3) ‘Only the smallest editing aspects in the text would have to be addressed.’
Response: We have made several language and editing corrections.
Reviewer 3 Report
This is an interesting study on the potential interplay between aldosterone and TGF-beta1 in cardiac hypertrophy and fibrosis. Overall, the study is designed and performed well, and the manuscript reads well. However, I have the following major concerns:
1) The proper loading controls for the immunoblots of p-ERK and p-p38 (Fig. 4) are total ERK and total p38 MAPK, respectively. Please repeat the western blotting with these loading controls.
2) In the same vein with my previous comment, why was RasGAP chosen as the loading control? There are a couple of issues with this: a) To my knowledge, there are several mammalian RasGAPs, so the authors need to specify which one the antibody they used recognized; and b) Especially in TGF-beta-induced hypertrophy, expression of RasGAPs could very well change (given that Ras regulates cell proliferation), so this disqualifies RasGAPs from being suitable as loading controls for the immunoblots of Fig. 4.
3) The finding that TGF-beta1 and aldosterone signaling pathways may converge during fibrosis (but not hypertrophy) induction is very interesting. Can the authors speculate on specific molecules these pathways may converge on? One such candidate protein could be the cardiac beta2-adrenergic receptor, which has been reported to modulate TGF-beta1 profibrotic signaling (PMID: 30897705), while also inhibiting aldosterone signaling via GRK5 activation (PMID: 32326036). Therefore, do the authors have any data on beta2-adrenergic receptor signaling, e.g., cAMP accumulation in response to isoproterenol or salbutamol, in their Ald-infused TGF-beta1-TG mice? Alternatively, they can just mention (briefly) this, along with other potential mechanisms, in "Discussion".
Minor English editing is warranted.
Author Response
Dear Reviewer 3,
we thank for your insightful remarks on our manuscript ‘Differential Role of Aldosterone and Transforming Growth Factor beta-1 in Cardiac Remodelling’.
We have been able to provide data of Western blots with total p38 and Erk as loading controls instead of RasGap (in fact most experiments were performed with both these proteins, i.e. p-Erk, erk and RasGap or p-p38, p38 and RasGap simultaneously).
Below we respond to your comments in a point-by-point manner below. The revised manuscript contains changes highlighted in yellow.
We hope you will find the corrected manuscript suitable for publication in the ‘International Journal of Molecular Sciences’.
Best regards,
Piotr Kmieć
Comment 1) The proper loading controls for the immunoblots of p-ERK and p-p38 (Fig. 4) are total ERK and total p38 MAPK, respectively. Please repeat the western blotting with these loading controls.
Response: In the revised manuscript, we were able to provide total Erk and total p38 as loading controls for p-Erk and p-p38, which was also accompanied by a new statistical analysis.
Comment 2a) (…) the authors need to specify which one the antibody they used recognized;
Response: In the revised manuscript, we have specified the antibody and provided an additional citation for reference.
Comment 2b) Especially in TGF-beta-induced hypertrophy, expression of RasGaps could very well change (…)
Response: We agree total Erk and total p38 are preferable loading controls for phosphorylated kinases. The revised manuscript contains modified Figure 4 and Results’ section, which takes into account a new analysis of Western blot data.
Comment 3) (…) do the authors have any data on beta2-adrenergic receptor signaling, e.g., cAMP accumulation in response to isoproterenol or salbutamol, in their Ald-infused TGF-beta1-TG mice? Alternatively, they can just mention (briefly) this, along with other potential mechanisms, in ‘Discussion’.
Response: Unfortunately, we did not carry out these experiments. TGF-beta1 transgenic mice were studied in this respect, however, without Ald infusion (Huntgeburth et al., 2011, PMID 22125598; Rosenkranz et al., 2002 cited in the manuscript).
In the revised manuscript, ‘Discussion’ was supplemented with several possible explanations for the convergence/absence of additive effects of TGF-beta1 and Ald on cardiac fibrosis.
Comment 4) Minor English editing is warranted.
Response: Language has been edited in the revised manuscript.
Round 2
Reviewer 1 Report
Accept
Author Response
Dear Reviewer,
thank you for endorsing the publication of our article.
Kind regards,
Piotr Kmieć
Reviewer 3 Report
I appreciate the effort by the authors to revise their manuscript according to my comments, most of which have been addressed adequately now. The revised manuscript is significantly improved. However, I still have a couple of minor concerns:
1) Although the correct loading controls for the western blots are shown now, it appears that total ERK (Fig. 4A) and total p38 (Fig. 4B & 4C) levels are quite different among samples, which makes no sense, unless the authors have evidence that total ERK and p38 protein expressions change in response to the various treatments. Is that what is happening here or do these differences merely reflect artifacts of their immunoblotting technique? If it`s the latter, they should use better representative blots to show in the main article, with equal protein levels of the loading controls (i.e., total ERK and total p38) shown.
2) In "Introduction", when discussing the molecular mechanisms of TGF-beta1-induced fibrosis, please mention also the important role of osteopontin, which is actually upregulated by aldosterone in cardiomyocytes (see: PMID: 30897705; a study that should be cited).
Minor editing is required.
Author Response
Comment 1) (...) it appears that total ERK (Fig. 4A) and total p38 (Fig. 4B & 4C) levels are quite different among samples (...) Is that what is happening here or do these differences merely reflect artifacts of their immunoblotting technique? (...)
Response: In Fig. 4A, total erk and total p38 are different between wild-type and transgenic mice, while in Fig. 4B-C of the first revision, loading inconsistensies are present.
In the second revision of the manuscript we have included RasGap for the WT vs TG comparison to indicate the differences in total erk and p38 amount. Also, we have provided different representative blots than previously (with more evenly loaded total Erk and total p38) in Figures 4B and 4C.
Unfortunately, we cannot provide better scans (neither for Fig. 4A nor B-C).
Comment 2: (...) please mention also the important role of osteopontin, which is actually upregulated by aldosterone in cardiomyocytes (see: PMID: 30897705; a study that should be cited).
Response: The requested publication has already been cited in the 'Discussion' in the first manuscript revision. In the newest version, conclusions of the study by Pollard et al. in that section have been rephrased. We would like to refrain from mentioning an in vitro study concerning osteopontin in a rather general 'Introduction'.